# Prevalence and Factors Associated with Insomnia in Military Personnel: A Retrospective Study during the Second COVID-19 Epidemic Wave in Peru

**DOI:** 10.3390/healthcare10071199

**Published:** 2022-06-27

**Authors:** Mario J. Valladares-Garrido, Cinthia Karina Picón-Reátegui, J. Pierre Zila-Velasque, Pamela Grados-Espinoza

**Affiliations:** 1Vicerrectorado de Investigación, Universidad Norbert Wiener, Lima 15046, Peru; 2Instituto de Evaluación de Tecnologías en Salud e Investigación-IETSI, EsSalud, Lima 15072, Peru; 3School of Medicine, Universidad de San Martín de Porres, Chiclayo 14012, Peru; cinthia_picon@usmp.pe; 4School of Medicine, Universidad Nacional Daniel Alcides Carrion, Pasco 19001, Peru; jzilav@undac.edu.pe (J.P.Z.-V.); pgradose@undac.edu.pe (P.G.-E.); 5Red Latinoamericana de Medicina en la Altitud e Investigación (REDLAMAI), Pasco 19001, Peru

**Keywords:** COVID-19, mental health, insomnia, public health, sleep quality

## Abstract

Studies in military personnel are scarce and have reported increased rates of medical consultations and insomnia. The COVID-19 pandemic has been associated with a number of factors that increase the prevalence of insomnia, which has established consequences in the military. However, reported data are from different settings. We aimed to identify the prevalence and factors associated with insomnia during the second COVID-19 epidemic wave in Lambayeque, Peru. A retrospective study in 566 participants was conducted face-to-face in November 2021. The dependent variable was insomnia, measured with the Insomnia Severity Index. The independent variables were socio-labor variables, physical activity, food insecurity, eating behavior disorder, fear of COVID-19, and resilience. The prevalence of insomnia was 23% (95% CI: 19.6–26.7%). In multivariate analysis, insomnia was associated with a personal history of mental health (PR: 1.71, 95% CI: 1.01–2.93), food insecurity (PR: 1.43, 95% CI: 1.05–1.95), fear of COVID-19 (PR: 2.57, 95% CI: 1.87–3.54), and high resilience (PR: 0.60, 95% CI: 0.42–0.86). Overall, the Peruvian military population presents a high prevalence of insomnia during the pandemic period. Special attention should be paid to factors that influence insomnia. Prevention and promotion programs should be established to reverse this negative trend in the military.

## 1. Introduction

At the end of 2019, the first case of COVID-19 infection was reported in China, Wuhan, which was declared a pandemic by the World Health Organization (WHO) on 11 March 2020. In Peru, up to 13,236 cases per day were reported, ranking as the fifth country with the highest mortality worldwide and confirming that the second wave of infection was devastating [1]. This situation led to changes in sleep patterns and rhythms across the board due to unprecedentedly high levels of stress [2]. Ideal sleep requires four essential attributes: good sleep quality, defined as the self-satisfaction of the sleep experience [3], an adequate sleep duration of at least 7 h, a regular circadian rhythm, and the absence of sleep disturbances [4]. Insomnia, defined as “difficulty initiating or maintaining sleep, waking too early, and inability to return to sleep,” is associated with waking fatigue [5].

The pandemic has variously affected the mental health of the general population. The highest estimate of depression was found in Africa (45%), of anxiety in Asia (34%), and of insomnia in Latin America (35%) [6]. Factors associated with insomnia in this context include death in the family, higher scores of stress, anxiety, and depression [7], having a job [8], having friends with COVID-19, and feeling estranged from family [9]. However, unlike the general population, military personnel represent a group that performs high-risk operations, such as emergency and disaster response. This involves intense exposure to stressors that lead to sleep disturbances, in addition to experiencing irregular and prolonged work schedules [10]. The American Academy of Sleep Medicine and the Sleep Research Society have proposed that adults get at least seven hours of sleep per night on a regular basis to maintain good health [11]. However, it is quite common for military personnel to get less than six hours of sleep per night [12], which translates into the development of sleep disorders.

Although research in military personnel is scarce, a study in the United States reported that the rates of consultations for insomnia increased from 16 to 75 per 1000 service members between 2005 and 2014 [13]. Another study in this country reported that insomnia rates increased from 6 to 272 per 10,000 service members between 2005 and 2019 [14]. Insomnia has also been identified as one of the most common symptoms of military personnel returning from deployment, with rates as high as 63.6% [15]. With respect to the consequences in the military, these have been reported to range from impairment of cognitive functions [16], increased risk of automobile accidents [17], and the development of chronic pathologies such as type II diabetes and hypertension [18], to a link with suicidal ideation [19].

The COVID-19 pandemic has affected mental health globally [20]. Although this context has been associated with the development of mental disorders, little research has been conducted in the military population. According to a pre-pandemic meta-analysis, insomnia is more likely to develop with age, alcohol dependence, white race, female sex, deployment and combat experience, depression, post-traumatic stress disorder, traumatic brain injury, and anxiety [21]. However, current studies have not evaluated other variables such as tobacco use, personal mental health history, work time, food insecurity, eating disorder, physical activity, and fear of COVID-19 infection. These variables are important because they may have a major influence on insomnia during the pandemic. In addition, previous studies in this context have included mostly military veterans [22], who represent a totally different group from the active military. Moreover, no reports of insomnia have been observed in military personnel from Latin American countries, in which the development of insomnia may occur differently.

Therefore, our study aimed to identify the prevalence and factors associated with insomnia in military personnel during the second wave of the COVID-19 epidemic in northern Peru.

## 2. Materials and Methods

### 2.1. Study Design and Population

A secondary data analysis of a cross-sectional study was conducted in military personnel in the city of Lambayeque, located in northern Peru. The data were collected from 2 to 9 November 2021, during the second COVID-19 epidemic wave in the country. In the primary study, the sample size was estimated considering a population size of 820 military personnel, an expected prevalence of 12.8% [23], a 99% confidence level, and a precision of 2.5%, which resulted in a sample size of 485 individuals. To this was added a 10% rejection rate and 10% incomplete registrations, calculating a required sample size of 582 individuals. Non-probabilistic snowball sampling was performed. Inclusion criteria were that the personnel were actively working during the pandemic or had worked at least 1 month. In the primary study, a total of 710 military personnel accepted to participate. For the present study, 144 of them were excluded as they did not fully respond to the Insomnia Severity Index (ISI). Therefore, the sample selected for the analysis consisted of 566 individuals.

We estimated a statistical power of 100% to evaluate the hypothesis of this study. We considered the expected prevalence of insomnia of 11.9% reported in Mexico [24] and the observed prevalence of insomnia of 23% according to the primary study, using a sample size of 582 participants.

### 2.2. Procedure

The present study was carried out in two phases: the first consisted of requesting permission from the Lambayeque military personnel infantry brigade and creating the questionnaire in the REDCap data entry system. The second phase consisted of face-to-face enrollment in compliance with biosecurity measures, which included data collection by field interviewers in two shifts lasting two hours. Finally, a member of the research team was instructed to carry out quality control of the data entered.

### 2.3. Questionnaire and Variables

The questionnaire consisted of 7 sections covering (1) sociodemographic data, (2) ISI, (3) resilience scale, (4) Household Food Security Access Scale (HFIAS), (5) Eating Disorder Scale (EAT-26), (6) Fear of COVID-19 Scale, and (7) Physical Activity Questionnaire (IPAQ-S).

General information was obtained on age (in years), sex (female, male), marital status (single, married, cohabitant, divorced), religion (none, Catholic, non-Catholic), self-report of frequent tobacco and alcohol consumption (no, yes), self-report of previous pathologies (arterial hypertension, type 2 diabetes), self-report of personal and family history of any mental health disorder (no, yes), seeking mental support (no, yes), having children (no, yes), trust in the government to manage the COVID-19 pandemic (no, yes), and time working in the military institution in the face of the COVID-19 pandemic (1 to 6 months, 7 to 12 months, 8 to 18 months, 19 months or more).

### 2.4. Dependent Variable

**Insomnia scale (ISI).** It is composed of 7 items that assess the nature, severity, and impact of insomnia. Higher scores reflect a higher degree of insomnia [25]. It has been validated in older adults, primary care patients, and the general Spanish-speaking population, demonstrating reliability with a Cronbach’s alpha of 0.82. [26]. The presence of insomnia was defined by a score of more than 8 points [27].

### 2.5. Independent Variables

**Resilience scale (CD-RISC).** The Connor-Davidson Short Resilience Scale, consisting of 10 items, was used. This instrument has been validated in the general Spanish-speaking population, showing a Cronbach’s alpha of 0.89 (general population) and a test–retest reliability of 0.87 (people with generalized anxiety disorder (GAD) and post-traumatic stress disorder) [28]. It was evaluated through a 5-point Likert scale with a score of 0 to 4. In general, it shows excellent psychometric properties and allows an efficient measurement of resilience [29]. A cut-off point of 30 was used to categorize high (>30) and low (<30) resilience [30].

**Scale of access to food security in the household (HFIAS).** This scale was developed by the United States Agency for International Development and includes 9 items, which correspond to questions about food in the last 4 weeks. The respondent is asked whether the household in which they live had experienced food insecurity in a given period, together with anxiety that they may have experienced, the quality and insufficient intake of food, and physical consequences. Responses are categorized into food insecurity (question 1), mild food insecurity (questions 2–4), moderate food insecurity (questions 5 and 6), and severe food insecurity (questions 6–9) [31]. The presence of food insecurity was categorized as mild, moderate, and severe. The instrument has been validated in Spanish-speaking older adults [31] and showed high internal consistency (α = 0.74) [32].

**Eating disorder scale (EAT-26).** This scale consists of 26 questions measured through a Likert scale with six response options (“never”, “rarely”, “sometimes”, “often”, “very often”, and “always”. The instrument has been validated in a Spanish-speaking female population, with a Cronbach’s alpha coefficient of 92.1% [33]. A score of 20 was considered the cut-off point for assuming the presence of an eating disorder [34].

**Physical activity questionnaire (IPAQ-S).** This scale considers the four components of physical activity (leisure, home maintenance, occupational, and transportation) [35]. It consists of 9 items and evaluates the physical activity reported in the last 7 days. It allows a weighted estimate of total physical activity to be obtained from the activities reported per week. The level of physical activity was categorized as low, moderate, and high. It has been validated in Spanish-speaking populations and applied in a Latin American population [36].

**Fear of COVID-19 Scale.** This scale consists of seven items and is reliable and valid for assessing fear of COVID-19 among the general population. It has shown a Cronbach’s alpha coefficient of 0.82 [37]. In the present study, a score greater than 16.5 was used as a cut-off point to assume the presence of fear of COVID-19 [38]. An investigation of the psychometric properties of the Spanish version of the COVID-19 Fear Scale in a sample of the Peruvian population demonstrated adequate measurement properties in terms of both reliability and validity [39].

### 2.6. Statistical Analysis

Survey data were organized in Microsoft Windows Excel^®^ and analyzed in Stata 16.1 (StataCorp LLC, College Station, TX, USA).

Descriptive statistics were used to identify sample characteristics. Categorical variables are described as frequencies and percentages, and continuous variables are described as mean (standard deviation) or median (range) values, as appropriate.

Results with *p*-values less than 0.05 were considered statistically significant. To evaluate the factors associated with the dependent variable (insomnia), we constructed simple and multiple regression models, estimated prevalence ratios (PRs) and 95% confidence intervals (95%CI), and used generalized linear models (GLM) with the Poisson distribution family, robust variance, and log-link function.

### 2.7. Ethical Considerations

The protocol of the present study was evaluated and approved by the Institutional Research Ethics Committee (CIEI) of the Universidad San Martin de Porres. Informed consent was obtained from each individual. Participation in the study was voluntary, without any form of coercion. The data were anonymous, coded, and confidential.

## 3. Results

The median age was 22 years, with an age range of 19 to 31 years. The male sex predominated (95.8%, *n* = 503). A total of 26.5% (*n* = 139) reported having children. In relation to substance use, alcoholism and smoking were present in 17.1% (*n* = 90) and 6.7% (*n* = 35) of respondents, respectively. In relation to comorbidities, overweight was present in 33.3% (*n* = 172), 9.5% (*n* = 50) had hypertension, and 1.9% (*n* = 10) had diabetes. Physical activity was high in 80.0% (*n* = 424), and working time was greater than 19 months in 36.3% (*n* = 186). Resilience was low in 56.4% (*n* = 296). Fear of COVID-19 was present in 19.2% (*n* = 101) of participants (Table 1). The prevalence of insomnia was 23% (95% CI: 19.56–26.66%). A detailed comparison of the severity of insomnia symptoms is shown in Figure 1.

In the simple regression analysis (Table 2), a statistically significant association was identified between insomnia and the presence of smoking (PR: 1.86, 95% CI: 1.09–3.19), personal history of mental health (PR: 3.19, 95% CI: 1.95–5.23), work time from 7 to 12 months (PR: 1.63, 95% CI: 1.03–2.55) and 8 to 18 months (PR: 1.74, 95% CI: 1.15–2.64), food insecurity (PR: 1.44, 95% CI: 1.06–1.96), a high level of physical activity (PR: 0.63, 95% CI: 0.43–0.92), eating behavior disorder (PR: 2.18, 95% CI: 1.55–3.07), fear of COVID-19 (PR: 3.36, 95% CI: 2.50–4.50), and high resilience (PR: 0.48, 95% CI: 0.34–0.69). Likewise, in the multiple regression analysis (Figure 2), the prevalence of insomnia was higher in people who have had a personal history of mental health (PR: 1.71, 95% CI: 1.01–2.93), food insecurity (PR: 1.43, 95% CI: 1.05–1.95), and fear of COVID-19 (PR: 2.57, CI: 1.87–3.54). In contrast, the prevalence of insomnia was reduced by 40% in military personnel with a high level of resilience (PR: 0.60, 95% CI: 0.42–0.86).

## 4. Discussion

### 4.1. Prevalence of Insomnia

The prevalence of insomnia in the military was 23% during the second wave of the COVID-19 epidemic in Peru. This result is higher than that found in the general Mexican population (11.9%) [24]. However, a higher prevalence of insomnia was found in the general Argentine population (65.6%) [40]. These differences could be due to the times at which the surveys were measured, the characteristics of the population, and the difference in the instruments used. This may also be different due to particular factors in the military, such as post-traumatic stress disorder, increased risk of physical injury, and experience of traumatic events [21]. In addition, our result is higher than those reported in U.S. military members, who showed a prevalence of 11.5%, 16.3%, and 19.9% in pre-deployment military, veterans, and active military from three major services, respectively [41,42,43]. These differences could be explained by the fact that previous studies were conducted in the United States, in which there has been better control of the pandemic than in Peru. However, no similar studies have been found in Latin America, so it is suggested to increase the regional evidence to determine measures to prevent insomnia in military personnel.

### 4.2. Factors Associated with Insomnia

In this study, having a history of mental health problems increased the prevalence of insomnia by 72%. This result coincides with a study that reported that having a history of head injury and mental health outcomes increases the probability of developing insomnia [43]. Likewise, a systematic review showed that the combined outcome of four comorbidities increases the risk in active duty personnel by up to 53% [21]. This finding could be explained by the fact that sleep disorders such as insomnia may be due to a psychiatric disorder and may also predispose a serving military member to develop post-traumatic stress disorder and other psychiatric conditions [44].

Having food insecurity increased the prevalence of insomnia by 43%. However, we did not find a similar study that associated these variables. Nevertheless, it differs from what was found in a study conducted in young people, in which 11% had food insecurity, and they were more likely to have problems falling asleep and staying asleep [45]. A similar situation was observed in a study that identified mental health outcomes and food insecurity in women, finding that food insecurity increased the risk of negative mental health symptoms [46]. The association that we found could be due to the fact that military personnel who have felt the effects of the economic impact of COVID-19 on their family basket probably experience excessive worry and high levels of stress, resulting in a lack of sleep [47].

Experiencing a high resilience pattern reduced the prevalence of insomnia by 40%. This situation supports the knowledge that attenuated stressor reactivity resulting from poor sleep quality reflects reduced resilience [48]. Our result is similar to a study that identified higher levels of insomnia in the group with low levels of resilience [43]. It should be noted that this study was conducted on a sample of military veterans. Likewise, the development of resilience as a measure of protection against mental health problems has been reported during the pandemic [22,49].

Fear of COVID-19 increased the prevalence of insomnia by 157%. This result is similar to that reported in Pakistani workers during the pandemic, in which fear of COVID-19 may be negatively associated with sleep quality [47]. This is similar to another study conducted in Bangladesh, in which fear of COVID-19 was found to be a significant predictor of sleep quality [50]. Our result coincides with a study conducted in university students that identified that the fear of contagion significantly influenced the development of insomnia in up to 32.9% [51]. It should be noted that these studies were not conducted in the military.

Military personnel in Peru have supported the containment of the epidemic waves of COVID-19 under the declaration of a national state of emergency. The measures applied consisted of supporting citizen security, which ensured compliance with the quarantine. Their participation contributed to reducing the number of COVID-19 infections and deaths [52]. However, greater exposure to pandemic-related events may have increased fear of contagion compared to the general population, contributing to increased rates of insomnia. We recommend that special attention be paid to potential predictors of insomnia in military personnel, the most important being the presence of an eating disorder and food insecurity. Appropriate agencies, through their welfare offices or mental health areas, should establish an ongoing assessment of the development of insomnia. In addition, adequate sleep quality programs should be implemented to reverse its consequences.

Our study has important strengths. We were able to obtain a large sample of the study population. Furthermore, the data were measured in person, which reduces measurement bias in the variables analyzed. Additionally, we used validated instruments to determine the severity of insomnia and other covariates influencing the dependent variable. However, the cross-sectional design of the primary study does not allow us to identify causal relationships between the study variables. In addition, it is not possible to generalize the results of this research to the entire population of interest, given that it captured data from military personnel from a single region of northern Peru. Finally, since the study is the result of a secondary data analysis, it was not possible to measure other important variables that may influence insomnia and sleep quality in the evaluated military personnel (e.g., post-traumatic stress disorder, combat and deployment experience, and number of traumatic events).

## 5. Conclusions

The prevalence of insomnia in Peruvian military personnel was higher than that in other military groups during the COVID-19 pandemic. However, more information is needed in this context to understand whether the pandemic significantly increased insomnia rates in the military. Special attention should be paid to the associated factors in order to focus interventions on vulnerable members. In addition, programs to promote good-quality sleep should be implemented. Our results contribute to the knowledge of insomnia in military personnel and to the potential implementation of policies to prevent its negative effects.

## Figures and Tables

**Figure 1 healthcare-10-01199-f001:**
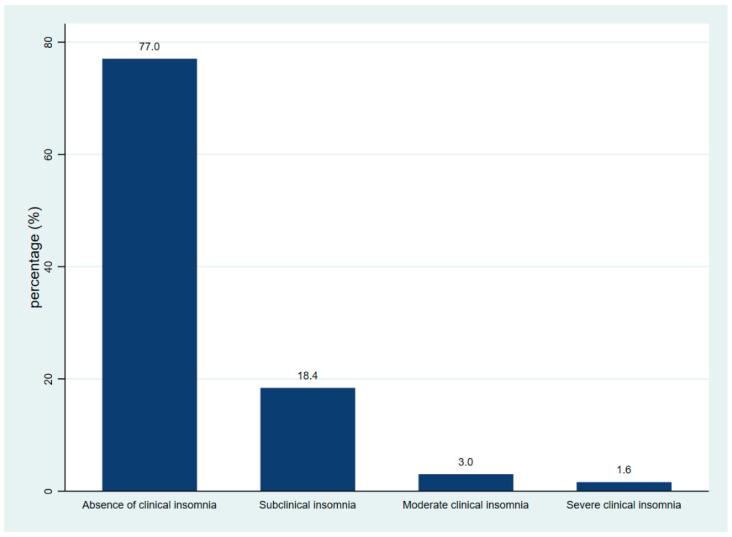
Frequency of insomnia symptoms according to severity based on Insomnia Severity Index-7.

**Figure 2 healthcare-10-01199-f002:**
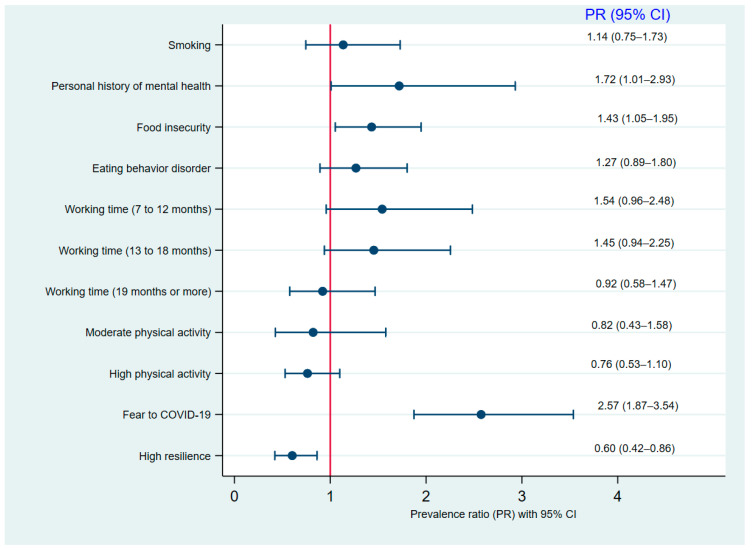
Forest plot of the factors associated with insomnia in multiple regression analysis.

**Table 1 healthcare-10-01199-t001:** Characteristics of study participants (*n* = 566).

Characteristics	*n* (%)
**Age**	22 (19–31) *
**Sex**	
Female	22 (4.2)
Male	503 (95.8)
**Marital status**	
Single	390 (74.3)
Married	117 (22.3)
Cohabitant	12 (2.3)
Divorced	6 (1.1)
**Religion**	
None	80 (15.2)
Catholic	359 (68.4)
Non-Catholic	86 (16.4)
**Parenting**	139 (26.5)
**Alcoholism**	90 (17.1)
**Smoking**	35 (6.7)
**Comorbidities**	
Hypertension	50 (9.5)
Diabetes	10 (1.9)
**Body mass index**	
Underweight/Normal	312 (60.4)
Overweight	172 (33.3)
Obesity	33 (6.4)
**Personal mental health history**	
No	518 (98.7)
Yes	7 (1.3)
**Family history of mental health**	
No	502 (95.6)
Yes	23 (4.4)
**Seeking mental health help**	
No	482 (91.8)
Yes	43 (8.2)
**Confidence in government to handle the pandemic**	
Yes	288 (54.9)
No	237 (45.1)
**Working time**	
1 to 6 months	134 (26.2)
7 to 12 months	82 (16.0)
13 to 18 months	110 (21.5)
19 months or more	186 (36.3)
**Food insecurity**	
No	265 (50.5)
Yes	260 (49.5)
**Physical activity**	
Low	64 (12.2)
Moderate	37 (7.1)
High	424 (80.8)
**Eating behavior disorder**	
No	471 (89.7)
Yes	54 (10.3)
**Resilience**	
Low	296 (56.4)
High	229 (43.6)
**Fear scale**	
No	424 (80.8)
Si	101 (19.2)
**Insomnia**	
Absence of clinical insomnia	436 (77.0)
Subclinical insomnia	104 (18.4)
Moderate clinical insomnia	17 (3.0)
Severe clinical insomnia	9 (1.6)

* Median (25–75th percentile).

**Table 2 healthcare-10-01199-t002:** Factors associated with insomnia in simple regression analysis.

Characteristics	Insomnia		
	Simple Regression		
	PR	95% CI	*p* *
**Age (years)**	0.99	0.97–1.00	0.146
**Sex**			
Female	Ref.		
Male	0.95	0.49–1.84	0.877
**Single**			
No	Ref.		
Yes	1.43	0.97–2.09	0.068
**Religion**			
None	Ref.		
Catholic	1.16	0.72–1.86	0.547
Non-Catholic	1.42	0.82–2.47	0.21
**Parenting**	0.72	0.49–1.04	0.08
**Alcoholism**	1.31	0.92–1.87	0.138
**Smoking**	1.86	1.09–3.19	**0.023**
**Comorbidities**			
Hypertension	1.26	0.79–2.00	0.319
Diabetes	1.6	0.72–3.56	0.247
**Body mass index**			
Underweight/Normal	Ref.		
Overweight	0.77	0.54–1.09	0.14
Obesity	1.07	0.61–1.89	0.81
**Personal history of mental health**			
No	Ref.		
Yes	3.19	1.95–5.23	**<0.001**
**Family history of mental health**			
No	Ref.		
Yes	1.23	0.64–2.35	0.529
**Confidence in government to handle the pandemic**			
Yes	Ref.		
No	1.17	0.86–1.58	0.312
**Working time**			
1 to 6 months	Ref.		
7 to 12 months	1.63	1.03–2.55	**0.035**
13 to 18 months	1.74	1.15–2.64	**0.009**
19 months or more	0.78	0.49–1.25	0.304
**Food insecurity**			
No	Ref.		
Yes	1.44	1.06–1.96	**0.019**
**Physical activity**			
Low	Ref.		
Moderate	0.6	0.29–1.21	0.151
High	0.63	0.43–0.92	**0.017**
**Eating behavior disorder**			
No	Ref.		
Yes	2.18	1.55–3.07	**<0.001**
**Fear of COVID-19**			
No	Ref.		
Yes	3.36	2.50–4.50	**<0.001**
**Resilience**			
Low	Ref.		
High	0.48	0.34–0.69	**<0.001**

* *p*-values obtained with generalized linear models, Poisson family, log-link function, and robust variance. Significant *p*-values are highlighted in bold.

## Data Availability

The dataset generated and analyzed during the current study is not publicly available because the ethics committee has not provided permission/authorization to publicly share the data but are available from the corresponding author on reasonable request.

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
