# Peer review of "Prevalence and Factors Associated with Insomnia in Military Personnel: A Retrospective Study during the Second COVID-19 Epidemic Wave in Peru"

_healthcare, 2022, doi:10.3390/healthcare10071199_

Round 1

Reviewer 1 Report

Valladares-Garrido et.al. present their study entitled: Prevalence and factors associated with insomnia in military 2 personnel: A cross-sectional study during the second COVID- 3 19 epidemic wave in Peru.

The whole article is well written, the topic is presented in an interesting way. However, I recommend that you slightly change the research results, especially their presentation. Currently, the results section contains mainly tables. It is also not entirely clear to potential interested in the topic. I recommend that you modify the results section and present some of them using graphs, other figures. It will definitely add value to your work. Good luck!

Reviewer 2 Report

The work presented is interesting because it concerns the military population. Generally, little international work concerns this population.

The results presented are interesting but in my opinion, could be improved. 

The introduction lacks some data on covid-19 and sleeps, so much literature has been published as this work

https://pubmed.ncbi.nlm.nih.gov/35023980/ 

but also many others.

In addition, it is not made explicit what kind of activities the military performed during the pandemic period. In many countries, they had specific activities for security and infection reduction.

However, the major criticism of the work is related to the type of statistical analysis performed. 

The first analysis compares the population by dividing insomnia and non-insomnia, and then a regression is performed. Both analyses actually bring the same results.

I believe that perhaps it would be more useful to try to enhance some of the variables examined and see their mediating role with respect to insomnia.

I suggest two works that have highlighted the role of resilience as a protective element for the population.

https://www.frontiersin.org/articles/10.3389/fpsyg.2021.646435/full

https://www.frontiersin.org/articles/10.3389/fpsyg.2020.567201/full

Or for example, one could look more into the psychological health aspect. Or again some analysis should be done on COVID, which actually emerges little in this work.

At present, the work as it stands does not add much information to understanding the phenomenon of insomnia.

Reviewer 3 Report

The manuscript presents a retrospective secondary data analysis to investigate factors related to insomnia among military personnel in the context of the Covid19 pandemic.

The study is interesting because it studies a population with special characteristics that may be underrepresented in other research. However, the data are not directly generalizable. There are several aspects that can be improved to make the final result better. 

The summary should be revised after the modifications proposed in the different sections.

Introduction: The characteristics of ideal sleep are presented and insomnia is defined. It then presents the particularities of sleep and insomnia among military personnel. It then discusses the impact of the Covid19 pandemic on mental health, and justifies the need for this research in that insomnia and its related factors have not been specifically studied in the active military population during the pandemic. Finally, the objective of the research is presented.

Aspects for improvement: In line 36 the quotation justifying the statement that "The attributes of ideal sleep cannot be met by the general population" is missing. It should be developed in detail what factors are related to insomnia in the general population, after the mentioned sentence indicating that it is difficult to achieve ideal sleep in that population. Then, the data mentioned in quotes 7 and 8 should be clarified. In quote 7, is the increase in consultations in absolute numbers or percentage? On what population? Is an increase from 16 to 75 relevant if these are absolute numbers? In quote 8, it should be stated where, in what context? The sentence in line 52 does not make sense when this manuscript has been carried out on a small population of military personnel that also does not allow generalizing the data. It is the same the text where quote 15 is mentioned. Variables that have not been taken into account in previous studies are mentioned (lines 57 to 60), but they are the ones that coincide later with those studied in this research. This could be provided that ALL known factors that may be related to insomnia have been previously developed in detail. It would be necessary to justify why the variables of the study are the most important or relevant taking into account all the known ones. It would also be necessary to develop in detail how the pandemic has affected the mental health of the general population, highlighting insomnia (if applicable), and then move on to the part of military personnel. 

Methodology: The location where the research was conducted, the study population and the sample are indicated. It is mentioned that this is a secondary analysis of data already existing in a primary study. The procedure, variables and validated questionnaires used in data collection are explained. The statistical analysis procedure is mentioned. The ethical principles of the research are complied with.

Aspects for improvement: The title states that it is a cross-sectional descriptive study, but since it is a secondary analysis, it is really a retrospective study, so the title should be modified. The date on which the data from the primary study were collected, which have been used here, should be indicated. The sample size does not agree with what has been explained. It is mentioned that in the primary study a sample of 582 participants was estimated, and that for this secondary analysis 144 people were excluded (582-144= 438), but it is said that the sample of the secondary analysis is 566 participants. It is necessary to indicate how the variables called general information (age, sex, marital status, etc...) have been categorized. There are some questionnaires that do not explain whether they have been validated in Spanish, unlike the others; this should be indicated (CD-RISC, HFIAS, EAT-26), and if they have not been validated in Spanish, what has been done (unvalidated version, English version, etc.). Although the ethical principles of research are formally complied with, since this is a research study with active military personnel, it is necessary to clarify how the situation was handled to avoid any doubt that there was some kind of order from the chain of command that could have influenced participation not being voluntary or free.

Results: The main results are presented in text and tables. 

Aspects for improvement: In tables 2 and 3, statistically significant p-values should be highlighted (in bold, or in italics, or with an asterisk...) to facilitate reading and interpretation. I do not see clearly the need to present bivariate analysis and simple regression. I would choose one of the two and then the multiple regression. But this is only an opinion. 

Discussion: The results regarding the prevalence of insomnia and the factors associated with insomnia are discussed. The results, similarities and differences with other studies, and possible reasons are discussed in detail. 

Aspects for improvement: The part on the prevalence of insomnia should be revised. It should be ordered by comparing it first with the general population and then with the military, and comment on whether the percentages are higher or lower and what may be the reason in each case. Factors that may cause more insomnia in military personnel have been mentioned in the introduction but are not now used in this part of the discussion to compare the prevalence data with the general population. In addition, variables that have not shown significance in multiple regression, even if they were significant in simple regression or bivariate analysis (eating disorder, smoking), should not be included in the discussion. Since they lost significance in the multiple regression, it cannot be affirmed that they are really associated with insomnia, but rather that the previous association was due to interactions of some kind with the other variables. Finally, given that the study was conducted during the pandemic, more weight should be given to the discussion of this factor, for example, by indicating what was happening at that time in the country that could have an impact on the fear of Covid (outbreaks, mortality, isolation measures, restrictions on movement, repercussions of all this on the military directly and on their families indirectly...). 

Conclusions: A summary conclusion of the data is mentioned and some conclusions are raised as a consequence of the results.

Aspects for improvement: It cannot be stated that the prevalence of insomnia is higher than in other contexts when in the discussion it is said that they are similar (citations 15 and 36), higher (citation 18 and 35) and lower (citation 34). The discussion and available data would need to be reviewed and that conclusion reworded. 

Round 2

Reviewer 2 Report

Thank you for accepting suggestion, I agree for publication

Reviewer 3 Report

Thank you for the effort to review the manuscript and incorporate the recommendations of the reviewers. In my opinion, all the proposed changes have been taken into account.I have no further suggestions for improvement.